# Phylomitogenomic Analyses Provided Further Evidence for the Resurrection of the Family Pseudoacanthocephalidae (Acanthocephala: Echinorhynchida)

**DOI:** 10.3390/ani13071256

**Published:** 2023-04-05

**Authors:** Tian-You Zhao, Rui-Jia Yang, Liang Lü, Si-Si Ru, Matthew Thomas Wayland, Hui-Xia Chen, Yuan-Hao Li, Liang Li

**Affiliations:** 1Hebei Key Laboratory of Animal Physiology, Biochemistry and Molecular Biology, Hebei Collaborative Innovation Center for Eco-Environment, College of Life Sciences, Hebei Normal University, Shijiazhuang 050024, China; 2Ministry of Education Key Laboratory of Molecular and Cellular Biology, Hebei Research Center of the Basic Discipline Cell Biology, Shijiazhuang 050024, China; 3Department of Zoology, University of Cambridge, Cambridge CB2 3EJ, UK

**Keywords:** Acanthocephala, Pseudoacanthocephalidae, *Pseudoacanthocephalus*, mitochondrial genome, phylogeny

## Abstract

**Simple Summary:**

Acanthocephalans, commonly known as spiny-headed or thorny-headed worms, are a small group of endoparasites with veterinary, medical and economic importance due to their ability to cause disease in domestic animals, wildlife, and humans. In recent decades, great progress has been made using mitochondrial genome data to clarify the phylogenetic relationships of acanthocephalans. However, the current mitochondrial genome database for acanthocephalans remains very limited. Herein, the characterization of the mitochondrial genome of *Pseudoacanthocephalus bufonis* (Shipley, 1903), the first representative of the family Pseudoacanthocephalidae, is reported. Phylogenetic analyses using the amino acid sequences of 12 protein-coding genes supported the validity of the family Pseudoacanthocephalidae and suggested a close affinity between Pseudoacanthocephalidae and Cavisomatidae. Our phylogenetic results also showed that the families Polymorphidae and Centrorhynchidae have a closer relationship than Plagiorhynchidae in the Polymorphida. These findings contribute to revealing the patterns of mitogenomic evolution in this group and represent a substantial step towards reconstructing the classification of the phylum Acanthocephala.

**Abstract:**

The phylum Acanthocephala is an important monophyletic group of parasites, with adults parasitic in the digestive tracts of all major vertebrate groups. Acanthocephalans are of veterinary, medical, and economic importance due to their ability to cause disease in domestic animals, wildlife, and humans. However, the current genetic data for acanthocephalans are sparse, both in terms of the proportion of taxa surveyed and the number of genes sequenced. Consequently, the basic molecular phylogenetic framework for the phylum is still incomplete. In the present study, we reported the first complete mitochondrial genome from a representative of the family Pseudoacanthocephalidae Petrochenko, 1956. The mitogenome of *Pseudoacanthocephalus bufonis* (Shipley, 1903) is 14,056 bp in length, contains 36 genes (12 protein-coding genes (PCGs) (lacking *atp8*), 22 tRNA genes, and 2 rRNA genes (*rrnL* and *rrnS*)) and two non-coding regions (*NCR1* and *NCR2*), and displayed the highest GC-skew in the order Echinorhynchida. Phylogenetic results of maximum likelihood (ML) and Bayesian inference (BI) using the amino acid sequences of 12 protein-coding genes in different models provided further evidence for the resurrection of the family Pseudoacanthocephalidae and also supported that the order Echinorhynchida is paraphyletic. A monophyletic clade comprising *P. bufonis* and *Cavisoma magnum* suggests a close affinity between Pseudoacanthocephalidae and Cavisomatidae. Our phylogenetic analyses also showed that Polymorphidae has a closer relationship with Centrorhynchidae than Plagiorhynchidae in the monophyletic order Polymorphida.

## 1. Introduction 

Acanthocephala is an important group of obligate endoparasites, with more than 1300 species parasitizing the digestive tracts of all major lineages of vertebrates and their larvae developing in arthropods [1,2,3,4]. According to the current classifications based on a combination of morphological and ecological traits, the phylum is divided into three classes, Archiacanthocephala, Eoacanthocephala, and Palaeacanthocephala, which include 10 orders, 26 families, and over 160 genera [1,5,6,7]. 

Some previous studies have made efforts to establish a basic molecular phylogenetic framework for Acanthocephala using various nuclear sequence data and mitochondrial genes [8,9,10,11,12,13,14,15,16,17]. Recently, mitochondrial genomic data were used to infer the phylogenetic relationships of the higher taxa in Acanthocephala [15,18,19,20,21,22,23,24,25]. However, to date, mitochondrial genome data are available for only 23 species of acanthocephalans, representing 13 families belonging to 6 orders. Several acanthocephalan families and orders were not represented in the above-mentioned phylogenetic studies, due to the paucity and inaccessibility of suitable material or genetic data for these groups. 

Echinorhynchida is the largest order in the phylum Acanthocephala, containing more than 470 nominal species, which mainly parasitize teleost fishes but also occur in amphibians and reptiles [1,5,26,27]. Amin (2013) listed 11 families in the Echinorhynchida, including Arhythmacanthidae, Cavisomatidae, Echinorhynchidae, Fessisentidae, Heteracanthocephalidae, llliosentidae, Isthmosacanthidae, Pomphorhynchidae, Rhadinorhynchidae, Transvenidae, and Sauracanthorhynchidae. Later, two new families, namely Gymnorhadinorhynchidae and Spinulacorpidae, were erected [6,28]. Additionally, two families, Paracanthocephalidae and Pseudoacanthocephalidae, were resurrected [16]. However, the phylogenetic relationships of these families are still uncertain. Additionally, the non-monophyly of Echinorhynchida was also revealed by some previous phylogenetic studies [8,9,10,11,12,13,14,17,18,29,30].

In order to further test the monophyly of the orders Echinorhynchida and Polymorphida, assess the validity of the recently resurrected family Pseudoacanthocephalidae, and clarify the evolutionary relationships of the Pseudoacanthocephalidae and the other families in Palaeacanthocephala using mitogenome data, the complete mitochondrial genome of *Pseudoacanthocephalus bufonis*, the first mitogenome from the Pseudoacanthocephalidae, was sequenced and annotated for the first time. Moreover, phylogenetic analyses of the protein-coding genes of all available acanthocephalan mitogenomes were performed using maximum likelihood (ML) and Bayesian inference (BI) in different models.

## 2. Materials and Methods 

### 2.1. Parasite Collection and Species Identification

A total of 14 spot-legged tree frogs, *Polypedates megacephalus* Hallowell (Anura: Rhacophoridae), were caught by hand at night in the Diaoluo Mountains, Hainan Island, China, and euthanized by injection of an overdose of pentobarbitone sodium solution. The acanthocephalan specimens were collected from the intestine of the host. For light microscopical studies, acanthocephalans were cleared in glycerine. Photomicrographs were recorded using a Nikon^®^ digital camera coupled to a Nikon^®^ optical microscopy (Nikon ECLIPSE Ni-U, Nikon Corporation, Tokyo, Japan). The specimens were identified as *P. bufonis* based on morphological features reported in previous studies [31,32,33,34,35,36]. The terminology is according to the previous study [37]. Voucher specimens were deposited in the College of Life Sciences, Hebei Normal University, Hebei Province, Shijiazhuang, China (HBNU-A-2022A001L).

### 2.2. Molecular Procedures

For molecular analysis, the genomic DNA was extracted using a modified CTAB (pH 8.0)-based DNA extraction protocol as described in Zhao et al. [38]. The genomic DNA library was constructed, and a total of 20 GB of clean data were generated using the pair-end 150 sequencing method on the Illumina NovaSeq 6000 platform by Novogene (Tianjin, China).

The complete mitochondrial genome was assembled using GetOrganelle v1.7.2a [39]. Protein coding genes (PCGs), rRNAs, and tRNAs were annotated using the MitoS web server (http://mitos2.bioinf.uni-leipzig.de/index.py, accessed on 20 January 2022) and MitoZ v2.4 [40]. The open reading frame (ORF) of each PCG was confirmed manually by the web version of ORF finder (https://www.ncbi.nlm.nih.gov/orffinder/, accessed on 10 March 2022). The “lost” tRNA genes ignored by both MitoS and MitoZ were identified using BLAST based on a database of the existing tRNA sequences of Acanthocephala. The secondary structures of tRNAs were predicted by the ViennaRNA module [41], building on MitoS2 [42] and RNAstructure v6.3 [43], followed by a manual correction. MitoZ v2.4 was used to visualize and depict gene element features [40]. The base composition, amino acid usage, and relative synonymous codon usage (RSCU) were calculated by a Python script, which refers to codon adaptation index (CAI) [44]. The total length of the base composition included ambiguous bases. The base skew analysis was used to describe the base composition of nucleotide sequences. The relative values were calculated using the formulas: $ATskew=\frac{A-T}{A+T}$ and $GCskew=\frac{G-C}{G+C}$. The complete mitochondrial genome sequence of *P. bufonis* obtained herein was deposited in the GenBank database (http://www.ncbi.nlm.nih.gov, accessed on 5 October 2022).

### 2.3. Phylogenetic Analyses

Phylogenetic analyses were performed based on concatenated amino acid sequences of 12 PCGs using maximum likelihood (ML) and Bayesian inference (BI). *Gnathostomula armata* and *G. paradoxa* (Gnathostomulida) were chosen as the out-group. The in-group included 7 species of rotifers and 24 species of acanthocephalans. Detailed information on representatives included in the present phylogeny was provided in Table 1. The phylogenetic trees were re-rooted on Gnathostomulida. Genes were aligned separately using MAFFT v7.313 under the iterative refinement method of E-INS-I [45]. Ambiguous sites and poorly aligned positions were pruned using BMGE v1.12 (m = BLOSUM90, h = 0.5) [46]. The aligned and pruned sequences were concatenated into a matrix by PhyloSuite v1.2.2 [47]. The pruned alignments were then concatenated into the “AA” matrix with the amino acid sequences of PCGs (2087 sites). Bayesian inference (BI) was implemented under the CAT + GTR + G4 model, using PhyloBayes-MPI 1.8 [45,48,49,50,51]. Two independent Markov Chain Monte Carlo (MCMC) runs of 8000 generations each were executed. A consensus tree was simultaneously built by pooling the remaining MCMC trees from both runs. Convergence was evaluated with the “bpcomp” and “tracecomp” procedures in the PhyloBayes package with a burn-in of the first 1000 generations. The maximum discrepancy in the convergence result is 0.017. The maximum likelihood (ML) inference was conducted in IQTREE v2.1.2 [52]. Substitution models were compared and selected according to the Bayesian Information Criterion (BIC) by using ModelFinder [53]. Additionally, a profile mixture model (C60) was used based on the best-fit substitution model of the NP datasets of amino acid datasets [54]. An edge-unlinked model was specified for both the full partition and the merged partition schemes. The best run is selected from the four independent runs based on log-likelihood. A total of 1000 Ultrafast bootstraps were used to evaluate the nodal support of the ML tree [55], and to estimate the consensus tree. For the “AA” matrix, three partition schemes were applied for ML (Table 2): (1) no partition (NP); (2) full partition (FP) that provides the best-fitting model for each individual gene; and (3) merged partition (MP) that implements a greedy strategy starting with the full partition model and subsequently merging pairs of genes until the model fit does not improve any further. The phylogenetic-terrace aware (PTA) data structure was used to facilitate the efficient analysis of the “AA” matrix under each partition model [56]. We selected the best final maximum likelihood and consensus trees according to the Akaike Information Criterion (AIC). Phylogenetic analyses ranked nodes with posterior probabilities (PP) and bootstrap support values (BS) = 1/100 as fully supported, 0.98–0.99/95–99 as strongly supported, 0.95–0.97/90–94 as generally supported, and <90/0.95 as weakly supported. The phylogenetic trees were visualized in iTOL v6.1.1 [57].

## 3. Results and Discussion

### 3.1. Morphology of Pseudoacanthocephalus bufonis (Figure 1, Table 3)

Trunks are medium-sized, smooth, and cylindrical. Females are much larger than males. Proboscis is nearly cylindrical, armed with 16–20 longitudinal rows of 3–5 rooted hooks each. Proboscis receptacle is double-walled with the cerebral ganglion at the posterior of the proboscis receptacle. The neck is short. Lemnisci are more or less equal, slightly longer or shorter than the proboscis receptacle. Morphometric data of the present specimens and morphometric comparisons of *P. bufonis* between our specimens and previous studies are shown in Table 3.

The morphology and measurements of the present material are more or less identical to the previous descriptions of *P. bufonis* [31,32,33,34,35,36], including the morphology and size of trunk and proboscis, the number of the longitudinal rows of proboscis hooks and the hooks per longitudinal row, the number and length of testes and cement-glands, and the morphology and size of eggs. However, the lengths of the proboscis receptacle and lemnisci are slightly ser than those of the previous studies. Additionally, we also sequenced the ITS region (OQ550505, OQ550506) of our specimens. Pairwise comparison of the ITS sequences of our specimens with the available ITS data (KC491878–KC491883) of *P. bufonis* reported in the previous study [70], displayed only 0.17% nucleotide divergence. Thus, we confirmed our specimens to be *P. bufonis*.

### 3.2. Gene Content and Organization of the Mitogenome

The complete mitogenome of *P*. *bufonis* is 14,056 bp in length and includes 36 genes, containing 12 PCGs (*cox1–3*, *nad1–6*, *nad4l*, *cytb,* and *atp6*), 22 tRNA genes, 2 rRNA genes (*rrnS* and *rrnL*), and two non-coding regions (*NCR1* and *NCR2*) (Figure 2, Table 4). The lack of *atp8* in the mitogenome of *P. bufonis* is typical for most of the available mitogenomes of acanthocephalans, except for *Leptorhynchoides thecatus*, which has two putative *atp8* genes [55]. All genes in the mitogenome of *P. bufonis* are encoded on the same strand and in the same direction. Furthermore, the highest GC-skew (0.53) and the second lowest AT-skew (−0.28) of the mitogenome of *P. bufonis* in the order Echinorhynchida show its preference for G and T nucleotides (Figure 3), which was possibly a result of the propensity for low use of A-rich codons in their PCGs (Table 5). A similar situation also occurred in *Polyacanthorhynchus caballeroi* and some species of Polymorphida [19,21,25].

### 3.3. Protein-Coding Genes and Codon Usage

The length of 12 PCGs is 10,114 bp. 12 PCGs encode 3358 amino acids and include 3358 codons, excluding termination codons. The longest PCG is *nad5* (1620 bp), while the shortest PCG is *nad4l* (243 bp) (Figure 2, Table 4).

The composition and usage of codons in the mitogenome of *P. bufonis* were shown in Figure 4 and Table 6. ATN (i.e., ATA, ATG, and ATT), GTG, and TTG are used as start codons for the 12 PCGs in the mitogenome of *P. bufonis*, whereas TAA, TAG, and incomplete codons of T or TA are used as termination codons, in accordance with those of other acanthocephalans [15,18,19,20,21,22,23,24,25,58,59,60,61,62,63,64]. GTG is the most common start codon, being used for six PCGs (*cox1*, *cox3*, *nad2*, *nad4l*, *nad5,* and *nad6*), followed by ATN for four PCGs (ATA: *atp6*; ATG: *nad3* and *nad4*; ATT: *nad1*). Two genes (*cytb* and *cox2*) were inferred to use TTG as the start codon. Among the 12 PCGs, six genes (*atp6*, *cytb*, *nad1*, *nad2*, *nad3,* and *nad4l*) are terminated with complete stop codon TAA, while three genes (*cox1*, *cox2,* and *nad5*) were inferred to terminate with complete stop codon TAG. The incomplete stop codons T and TA are used for *cox3* and *nad4*, and *nad6*, respectively.

In the PCGs of *P. bufonis*, the codon with the highest RSCU value is AGG (Ser), while the rarest codon is CTC (Leu). Val is the most frequently used amino acid (16.66%) in 12 PCGs of *P. bufonis*. Gln is the least commonly used amino acid (0.59%). The high frequency of Val (encoded by GTN) is associated with the high proportions of G and T in their protein-coding sequences (Figure 4 and Table 6).

### 3.4. Ribosomal and Transfer RNAs

A total of 22 tRNAs were identified, ranging in length from 42 bp (*trnC*) to 68 bp (*trnI*) (Table 4). The anticodons (Table 4) and secondary structures (Figure 5) of the 22 tRNAs were identified. Of the 22 tRNAs, four (*trnA*, *trnN*, *trnL2*, *trnS1*) have a short dihydrouridine (DHU) arm, six (*trnA*, *trnE*, *trnG*, *trnI*, *trnL1*, *trnY*) lack a TψC (T) arm, and two (*trnD* and *trnC*) have lost both arms (Figure 5). Moreover, the *trnT* has a short amino acid acceptor (AA) arm. The other nine tRNAs were predicted to be folded into typical cloverleaf secondary structures, as found in other acanthocephalans [19,23].

In the mitogenome of *P. bufonis*, two rRNAs, *rrnL,* located between *trnY* and *trnL1*, and *rrnS,* located between *trnM* and *trnP*, were identified. The *rrnL* is 908 bp in length, with 62.86% A + T content, whereas the *rrnS* is 572 bp in length with 64.37% A + T content (Figure 2 and Table 5).

### 3.5. Gene Order

In the mitogenome of *P. bufonis*, gene arrangement of PCGs and rRNAs is in the following order: *cox1*, *rrnL*, *nad6*, *atp6*, *nad3*, *nad4l*, *nad4*, *nad5*, *ctyb*, *nad1*, *rrnS*, *cox2*, *cox3,* and *nad1*, a pattern which appears to be relatively conserved in acanthocephalans [22,23,24,25,64]. However, some tRNAs (i.e., *trnS1*, *trnS2*, *trnM*, *trnV*, *trnK*, *trnR,* and *trnC*) show more variability in translocation [21,65]. There are up to three translocations in tRNAs in the mitogenome of acanthocephalans reported so far (i.e., *trnS1*, *trnS2*, and *trnK*) [18,20,22,23,24,66] (Figure 6). There are two main gene arrangements of *trnS2*: type A (*trnS2*, *atp6*, *nad3*, *trnW*, *trnV*, *trnK*, *trnE*, and *trnT*) and type B (*atp6*, *nad3*, *trnW*, *trnV*, *trnK*, *trnE*, *trnT*, and *trnS2*). The *trnK* has two arrangements: type C (*trnK* and *trnV*) and type D (*trnV* and *trnK*). The gene arrangement of *trnS1* has three order types: type E (*trnS1*, *trnM*, *rrnS*, *trnF*, *cox2*, *trnC*, *cox3*, *trnA*, *trnR*, and *trnN*), type F (*trnM*, *rrnS*, *trnF*, *cox2*, *trnC*, *cox3*, *trnA*, *trnR*, *trnN*, and *trnS1*), and type G (*trnM*, *trnS1*, *rrnS*, *trnF*, *cox2*, *trnC*, *cox3*, *trnA*, *trnR*, and *trnN*) [18,20,22,23,24,59,66,67]. In the mito-genome of *P. bufonis*, *trnR* is situated between *trnA* and *trnN*, while *trnS2* lays between *trnD* and *atp6* (Figure 6).

The gene order of *trnS2* in the mitogenome of *P. bufonis* is type A (*trnS2*, atp6, *nad3*, *trnW*, *trnV*, *trnK*, *trnE*, *trnT*). The gene arrangement of *trnK* is of type D (*trnK*, *trnV*). The *trnS1* of *P. bufonis* is of type F (*trnM*, *rrnS*, *trnF*, *cox2*, *trnC*, *cox3*, *trnA*, *trnR*, *trnN*, *trnS1*) (Figure 6).

### 3.6. Non-Coding Regions

In the mitogenome of *P. bufonis*, there are two non-coding regions (*NCR1* and *NCR2*). *NCR1* is located between *trnI* and *trnM*, is 610 bp in length. *NCR2,* located between *trnW* and *trnV*, is 503 bp. Their A + T contents are 61.97% and 56.86%, respectively (Table 5).

### 3.7. Molecular Phylogeny

Phylogenetic trees generated from BI and ML methods under different models have similar topologies and indicate that the Acanthocephala are monophyletic, which was widely accepted in previous studies. However, the evolutionary relationships of the Acanthocephala and the three subtaxa of Rotifera (Monogononta, Bdelloidea, and Seisonidea) have been under debate for a long time [19,71,72,73]. The present phylogenetic results showed that the Acanthocephala is sister to Bdelloidea (*Rotaria rotatoria*, *Philodina citrina*) and rejected the monophyly of Eurotatoria (Monogononta + Bdelloidea), which are identical to the previous phylogenetic results using EST libraries [72] and mitogenomic data [19], but conflicted with some other phylogenies based on 18S rDNA and transcriptomic data [71,73].

Our phylogeny also supported the division of the phylum Acanthocephala into three large clades (Clade I, Clade II, and Clade III) (Figure 7). Clade I, including *Macracanthorhynchus hirudinaceus* and *Oncicola luehei* (Oligacanthorhynchida: Oligacanthorhynchidae), represents Archiacanthocephala, a monophyletic group located at the base of the phylogenetic trees of Acanthocephala (Figure 7). The present results agree well with some previous phylogenetic studies [8,9,11,12,13,18,20,22,23,24,25,64,66]. The representative of Polyacanthocephala (*Polyacanthorhynchus caballeroi*) nested with species of Eoacanthocephala (*Pallisentis celatus* + *Acanthogyrus bilaspurensis* + *Neoechinorhynchus violentum* + *Paratenuisentis ambiguus*), forming Clade II. The present phylogenetic results challenged the validity of Polyacanthocephala, as have some previous molecular phylogenetic studies [19,22,23,25,64,66].

The representatives of Palaeacanthocephala formed Clade III. The monophyly of the order Polymorphida, including the representatives of *Plagiorhynchus transversus*, *Polymorphus minutus*, *Southwellina hispida*, *Centrorhynchus clitorideus*, *C. milvus*, *C. aluconis*, *Sphaerirostris lanceoides,* and *S. picae*, is strongly supported in our phylogenetic results. In Polymorphida, the Polymorphidae are more closely related to the Centrorhynchidae than the Plagiorhynchidae, in accordance with other recent mitogenomic phylogenies, but inconsistent with some previous phylogenetic studies based on nuclear and mitochondrial genetic markers [17,68,69,74]. Our phylogenetic results showed that the order Echinorhynchida is paraphyletic, which is consistent with previous molecular phylogenetic studies [12,14,28,30,69]. Furthermore, they supported the resurrection of Pseudoacanthocephalidae [16]. In the present mitogenomic phylogeny, *P. bufonis* clustered together with *Cavisoma magnum*, suggesting an affinity between Pseudoacanthocephalidae and Cavisomatidae. Our results agreed well with some recent phylogenetic studies based on nuclear gene sequences [16,75]. These indicated that the current classification of Echinorhynchida is based on unique combinations of characteristics, not shared derived features [13]. The systematics of Echinorhynchida needs to be revised so that its constituent families, subfamilies, and genera reflect the underlying lineages. This will require phylogenetic analysis of both nuclear and mitochondrial DNA datasets from representatives of a more diverse range of taxa than are currently available. It is essential to sequence mitogenomes from yet unrepresented taxa for constructing the molecular phylogenetic framework of Acanthocephala and further exploring the unusual patterns of mitogenomic evolution in this group. The complete mitogenome of *P. bufonis* obtained herein represents a valuable building block for future work.

## 4. Conclusions

In the present study, the complete mitochondrial genome of *P. bufonis*, the first representative of the family Pseudoacanthocephalidae, was characterized. Phylogenetic analyses based on the amino acid sequences of 12 protein-coding genes further confirmed the sister relationship of the Acanthocephala and Bdelloidea and rejected the monophyly of Eurotatoria (Monogononta + Bdelloidea) and Pararotatoria (Seisonidea + Acanthocephala). Our phylogeny also revealed that the order Echinorhynchida and the family Echinorhynchidae are both paraphyletic in the Acanthocephala. The current systematic status of *Pseudoacanthocephalus* in the Echinorhynchidae is challenged. The present phylogenetic results supported the recent resurrection of Pseudoacanthocephalidae and showed a close affinity between Pseudoacanthocephalidae and Cavisomatidae. Phylogenetic analyses also strongly supported the monophyly of the order Polymorphida and indicated that the Polymorphidae and Centrorhynchidae have a closer relationship than the Plagiorhynchidae. The present phylogenetic studies provided a new insight into the evolutionary relationships of higher taxa within Acanthocephala.

## Figures and Tables

**Figure 1 animals-13-01256-f001:**
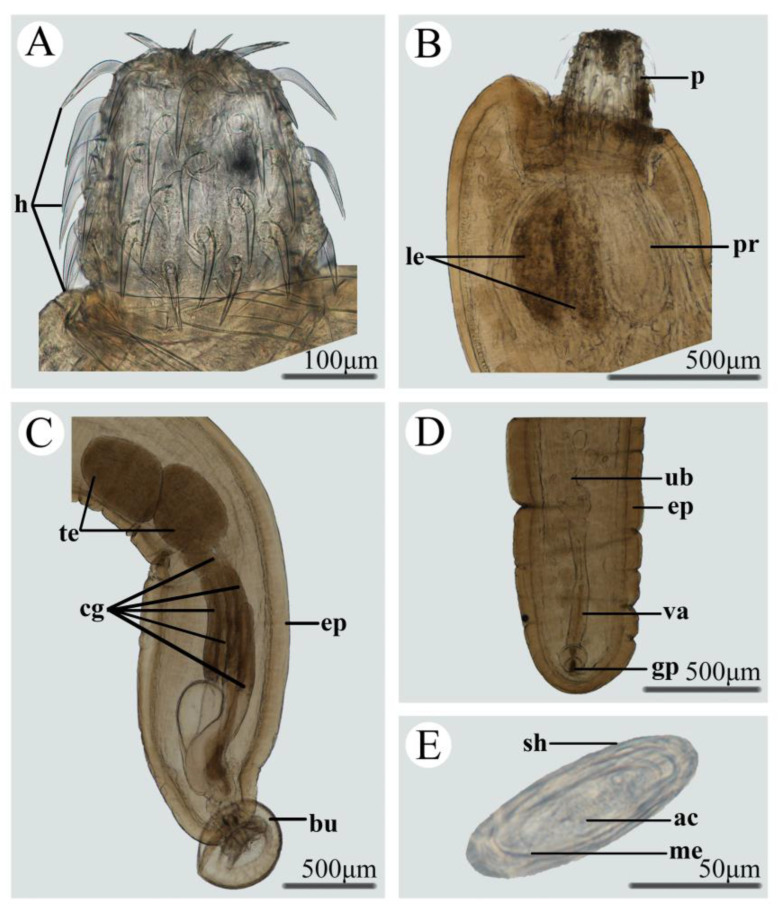
Photomicrographs of *Pseudoacanthocephalus bufonis*. (**A**) Proboscis; (**B**) Presoma of female; (**C**) Metasoma of male; (**D**) Metasoma of female; (**E**) egg. Abbreviations: ac, acanthor; bu, copulatory bursa; cg, cement glands; ep, epidermis; h, hooks; le, lemniscs; me, membrane; p, proboscis; pr, proboscis receptacle; sh, shell of egg, te, testis; ub, uterine bell; va, vagina.

**Figure 2 animals-13-01256-f002:**
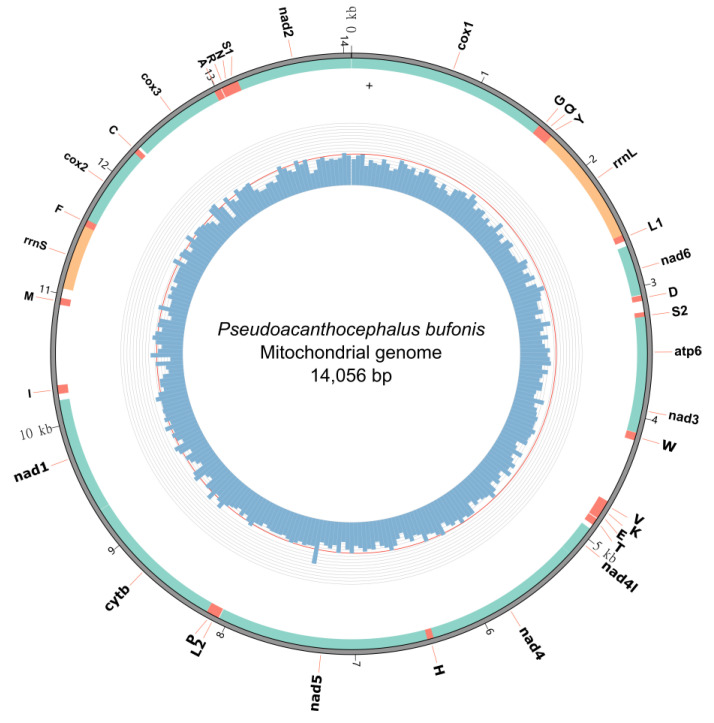
Gene map of the mitochondrial genome of *Pseudoacanthocephalus bufonis*. The outermost circle shows the gene features, sandy brown for rRNAs, salmon for tRNAs, and light sea green for PCGs. The innermost circle shows the GC content calculated in every 50-site window.

**Figure 3 animals-13-01256-f003:**
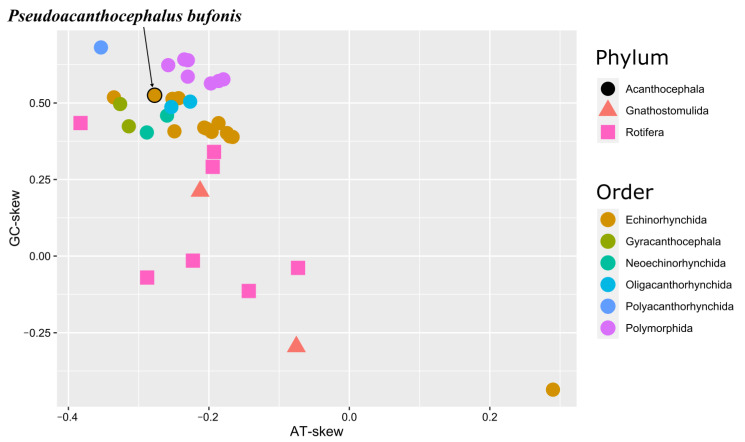
Comparison of genome-wide nucleotide skews among Acanthocephala, Gnathostomulida, and Rotifera. Species of Acanthocephala are coloured according to their taxonomic placement at the Order level.

**Figure 4 animals-13-01256-f004:**
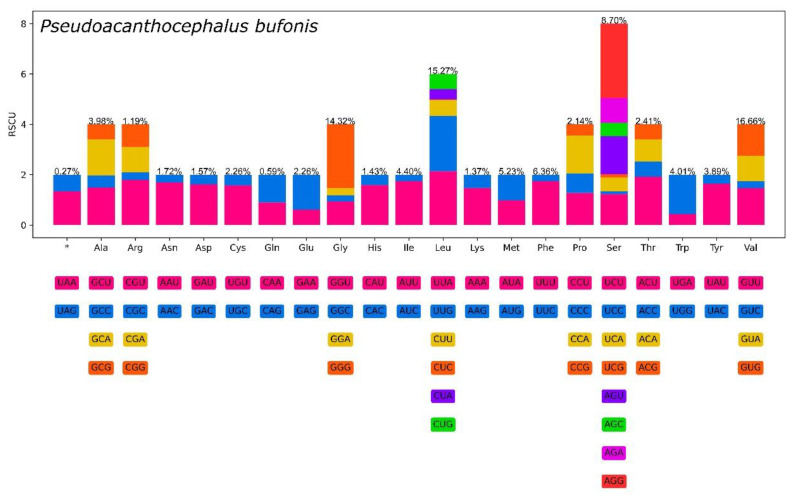
Relative synonymous codon usage (RSCU) of *Pseudoacanthocephalus bufonis*. Codon families (in alphabetical order) are provided below the horizontal axis.

**Figure 5 animals-13-01256-f005:**
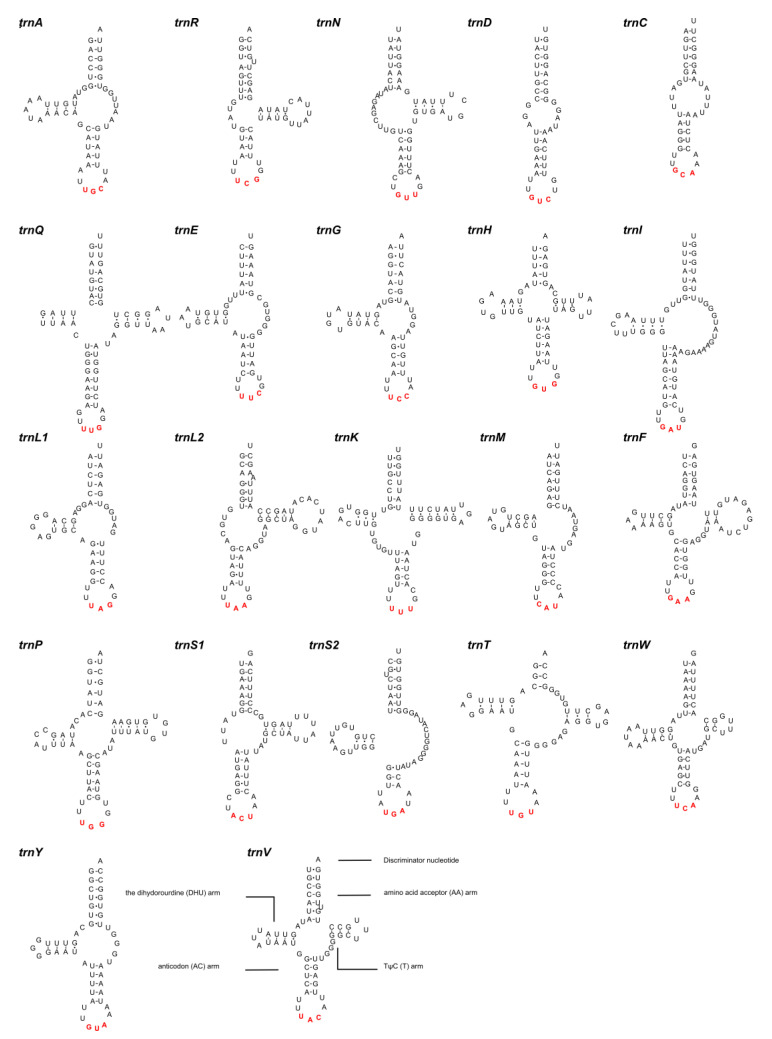
Predicted secondary structures of 22 tRNAs in the mitogenome of *Pseudoacanthocephalus bufonis* (Watson-Crick bonds indicated by lines, GU bonds indicated by dots, red bases representing anticodons).

**Figure 6 animals-13-01256-f006:**
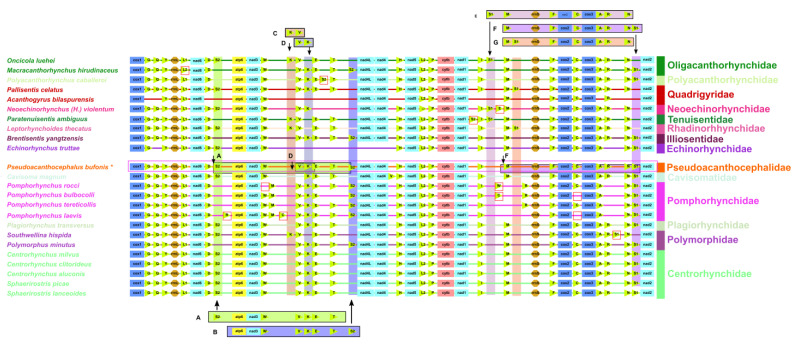
Comparison of the linearized mitochondrial genome arrangement for 24 acanthocephalans species. All genes are transcribed in the same direction, from left to right. The tRNAs are labelled by a single-letter code for the corresponding amino acid. A–B: the position of *trnS2*; C–D: the position of *trnK*; E–G: the position of *trnS1*. Translocations of individual *tRNAs* are marked with red boxes. The shadowed regions highlighted bin pink represent Pomphorhynchidae (*Pseudoacanthocephalus bufonis* indicated by asterisk).

**Figure 7 animals-13-01256-f007:**
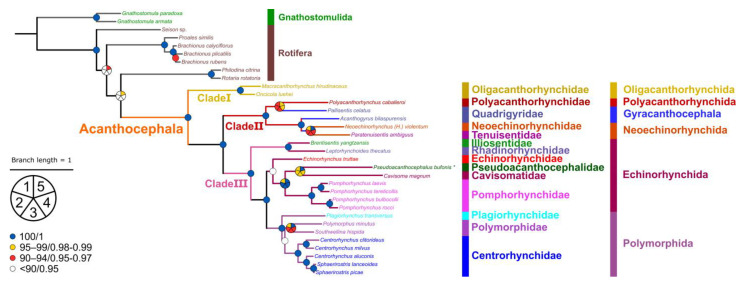
Phylogenetic relationships of Acanthocephala are presented based on the topology of Bayesian inference results. The branch length scale is marked according to the tree of Bayesian inference. The species followed by an asterisk (*) are sequenced in this study. Coloured circles indicate posterior probabilities/bootstrap support. Nodes of the cladogram with 0.95–0.97/90–94 are labelled orange; 0.98–0.99/95–99 by blue; 1/100 by green. Nodes divided into five parts indicate the different supports of five methods: (1) Bootstrap support of ML methods with no partition (NP) scheme under the mtZOA + F + R5 model; (2) with full partition (FP) scheme; (3) with merged partition (MP) scheme; (4): Bootstrap support of ML methods under the mtZOA + F + R5 + C60 model; and (5) Posterior probabilities of Bayesian inference.

**Table 1 animals-13-01256-t001:** Detailed information on representatives included in the present phylogeny.

Phylum/Class	Order	Family	Species	Accession	Length	AT%	References
Outgroup							
Gnathostomulida	Bursovaginoidea	Gnathostomulidae	*Gnathostomula armata*	NC_026983	14,030	80.2	[55]
			*Gnathostomula paradoxa*	NC_026984	14,197	71.8	[55]
Ingroup							
Rotifera	Bdelloidea	Philodinidae	*Rotaria rotatoria*	NC_013568	15,319	73.2	[58]
			*Philodina citrina*	FR856884	14,003	77.7	[20]
	Monogononta	Brachionidae	*Brachionus calyciflorus*	KX822781	27,683	68.7	[59]
			*Brachionus plicatilis*	NC_010484	12,672	62.9	[60]
			*Brachionus rubens*	MN256532	13,795	67.2	[61]
		Proalidae	*Proales similis*	MN970216	16,819	67.2	[62]
	Seisonidea	Seisonidae	*Seison* sp.	KP742964	15,120	70.0	[63]
Acanthocephala							
Archiacanthocephala	Oligacanthorhynchida	Oligacanthorhynchidae	*Macracanthorhynchus hirudinaceus*	NC_019808	14,282	65.2	[20]
			*Oncicola luehei*	NC_016754	14,281	60.2	[15]
Eoacanthocephala	Gyracanthocephala	Quadrigyridae	*Pallisentis celatus*	NC_022921	13,855	61.5	[64]
			*Acanthogyrus bilaspurensis*	MT476589	13,360	59.3	unpublished
	Neoechinorhynchida	Neoechinorhynchidae	*Neoechinorhynchus violentum*	KC415004	13,393	59.4	[65]
		Tenuisentidae	*Paratenuisentis ambiguus*	NC_019807	13,574	66.9	[20]
Palaeacanthocephala	Echinorhynchida	Cavisomidae	*Cavisoma magnum*	MN562586	13,594	63.0	[24]
		Echinorhynchidae	*Echinorhynchus truttae*	NC_019805	13,659	63.1	[20]
		Pseudoacanthocephalidae	*Pseudoacanthocephalus bufonis*	MZ958236	14,056	58.4	present study
		Illiosentidae	*Brentisentis yangtzensis*	MK651258	13,864	68.3	[66]
		Pomphorhynchidae	*Pomphorhynchus bulbocolli*	JQ824371	13,915	59.9	unpublished
			*Pomphorhynchus laevis*	JQ809446	13,889	57.1	unpublished
			*Pomphorhynchus rocci*	JQ824373	13,845	60.7	unpublished
			*Pomphorhynchus tereticollis*	JQ809451	13,965	56.9	unpublished
		Rhadinorhynchidae	*Leptorhynchoides thecatus*	NC_006892	13,888	71.4	[67]
	Polymorphida	Centrorhynchidae	*Centrorhynchus clitorideus*	MT113355	15,884	55.5	[68]
			*Centrorhynchus milvus*	MK922344	14,314	54.5	[22]
			*Centrorhynchus aluconis*	KT592357	15144	54.5	[19]
			*Sphaerirostris lanceoides*	MT476588	13,478	58.0	[25]
			*Sphaerirostris picae*	MK471355	15,170	58.1	[23]
		Polymorphidae	*Polymorphus minutus*	MN646175	14,149	64.4	[69]
			*Southwellina hispida*	NC_026516	14,742	63.9	[18]
		Plagiorhynchidae	*Plagiorhynchus transversus*	NC_029767	15,477	61.1	[19]
Polyacanthocephala	Polyacanthorhynchida	Polyacanthorhynchidae	*Polyacanthorhynchus caballeroi*	NC_029766	13,956	56.3	[19]

**Table 2 animals-13-01256-t002:** Models, partitioning schemes, and model comparisons of the maximum-likelihood analyses. t: number of partitions; k: number of free parameters; ln (Lik): log-likelihood; AIC: Akaike information criterion; ΔAIC: difference from the minimum AIC; BIC: Bayesian information criterion NP: unpartitioned model; FP: edge-unlinked full partition model; MP: merged and edge-unlinked partition model.

Partition Scheme (t)	Model	Parameters (k)	ln (Lik)	AIC	ΔAIC	BIC
NP (1)	mtZOA + F + R5 + C60	150	−64,770.62	129,841.23		130,683.08
MP (2)	-	112	−64,937.53	130,099.06	257.83	130,727.65
NP (1)	mtZOA + F + R5	90	−65,020.09	130,220.18	378.94	130,725.29
FP (12)	-	287	−64,983.58	130,541.16	699.92	132,151.90

**Table 3 animals-13-01256-t003:** Comparative morphometric data for *Pseudoacanthocephalus bufonis* (all measurements are in millimetres).

Host	*Polypedates megacephalus*		*Bufo melanostictus, Rana cancrivora, Takydromus sexlineatus*		*Bufo melanostictus*		*Polypedates megacephalus, P. mutus, Fejervarya limnocharis, Limnonectes kuhlii, Philautus odontotarsus, Odorrana versabilis, Rana livida, Amolops* sp.	
Locality	China		Indonesia		China		China	
Source	Present study		Kennedy (1982) [32]		Wang (1989) [33]		Bush (2009) [34]	
Characteristics	Male	Female	Male	Female	Male	Female	Male	Female
Trunk length	5.43–8.98	10.0–18.0	5.29–9.38	11.2–16.1	6.80–8.00	15.0–20.0	5.70–11.8	15.3–28.0
Trunk width	1.00–1.50	0.98–1.45	0.98–1.51	1.13–2.10	0.88–1.52	1.40–1.60	1.00–2.20	1.10–2.30
Proboscis length	0.31–0.50	0.38–0.53	0.31–0.44	0.33–0.54	0.48–0.52	0.56–0.64	0.41–0.54	0.41–0.54
Proboscis width	0.28–0.31	0.30–0.41	0.17–0.31	0.22–0.33	0.32–0.45	0.32–0.46	0.28–0.36	0.28–0.36
Lemnisci length	0.48–0.95	0.59–1.23	0.74–1.32	1.16–1.70	0.96–1.44	0.72–1.44	0.80–1.42	0.80–1.42
Proboscis receptacle length	0.41–0.71	0.58–0.91	0.65–0.90	0.88–1.10	0.96–1.12	0.88–1.12	0.69–0.95	0.69–0.95
Proboscis receptacle width	0.25–0.30	0.25–0.45	0.19–0.35	0.31–0.53	0.21–0.32	0.24–0.32	0.27–0.35	0.27–0.35
Size of the anterior testis	0.45–0.80	N/A	0.47–0.79	N/A	0.56–0.68	N/A	0.51–0.95	N/A
	× 0.40–0.62		× 0.31–0.53		× 0.40–0.48		× 0.29–0.58	
Size of the posterior testis	0.45–0.74	N/A	0.54–0.72	N/A	0.52–0.72	N/A	0.51–0.95	N/A
	× 0.37–0.65		× 0.28–0.50		× 0.48–0.51		× 0.29–0.58	
Cement-gland length	0.76–1.46	N/A	–	N/A	–	N/A	0.77–1.51	N/A
Size of the copulatory bursa	0.40–0.46	N/A	–	N/A	–	N/A	–	N/A
	× 0.46–0.84							
Uterine bell length	N/A	0.34–0.62	N/A	–	N/A	0.80–0.85	N/A	0.45–0.68
Uterus length	N/A	0.26–0.47	N/A	–	N/A	–	N/A	0.29–0.38
Size of the egg	N/A	0.06–0.09	N/A	0.08–0.09	N/A	0.06–0.09	N/A	0.06–0.07
		× 0.02–0.03		× 0.02–0.03		× 0.02–0.03		× 0.02

**Table 4 animals-13-01256-t004:** Annotations and gene organization of *Pseudoacanthocephalus bufonis*. The positive number in the “Gap or overlap” column indicates the length of intergenic sequence, and the negative number indicates the length (absolute number) that adjacent genes overlap (negative sign).

Gene	Type	Start	End	Length	Start Codon	Stop Codon	Anticodon	Gap or Overlap
*cox1*	CDS	1	1539	1539	GTG	TAG		−2
*trnG*	tRNA	1538	1589	52			UCC	−11
*trnQ*	tRNA	1579	1641	63			UUG	−13
*trnY*	tRNA	1629	1679	51			GUA	1
*rrnL*	rRNA	1681	2588	908				0
*trnL1*	tRNA	2589	2638	50			UAG	0
*nad6*	CDS	2639	3063	425	GTG	TA		4
*trnD*	tRNA	3068	3111	44			GUC	85
*trnS2*	tRNA	3197	3253	57			UGA	−22
*atp3*	CDS	3232	3771	540	ATA	TAA		−1
*nad3*	CDS	3771	4118	348	ATG	TAA		1
*trnW*	tRNA	4120	4179	60			UCA	0
*NCR2*	Non-coding region	4180	4682	503				0
*trnV*	tRNA	4683	4742	60			UAC	−14
*trnK*	tRNA	4729	4793	65			UUU	−11
*trnE*	tRNA	4783	4833	51			UUC	5
*trnT*	tRNA	4839	4894	56			UGU	35
*nad4l*	CDS	4930	5172	243	GTG	TAA		0
*nad4*	CDS	5173	6397	1225	ATG	T		1
*trnH*	tRNA	6399	6453	55			GUG	−6
*nad5*	CDS	6448	8067	1620	GTG	TAG		4
*trnL2*	tRNA	8072	8133	62			UAA	−23
*trnP*	tRNA	8111	8173	63			UGG	0
*cytb*	CDS	8174	9298	1125	TTG	TAA		1
*nad1*	CDS	9300	10,190	891	ATT	TAA		47
*trnI*	tRNA	10,238	10,305	68			GAU	0
*NCR1*	Non-coding region	10,306	10,915	610				0
*trnM*	tRNA	10,916	10,969	54			CAU	0
*rrnS*	rRNA	10,970	11,541	572				0
*trnF*	tRNA	11,542	11,605	64			GAA	8
*cox2*	CDS	11,614	12,216	603	TTG	TAG		−2
*trnC*	tRNA	12,215	12,256	42			GCA	19
*cox3*	CDS	12,276	12,978	703	GTG	T		0
*trnA*	tRNA	12,979	13,032	54			UGC	5
*trnR*	tRNA	13,038	13,089	52			UCG	−20
*trnN*	tRNA	13,070	13,127	58			GUU	−10
*trnS1*	tRNA	13,118	13,174	57			ACU	−1
*nad2*	CDS	13,175	14,055	882	GTG	TAA		1

**Table 5 animals-13-01256-t005:** Base composition and skewness of *Pseudoacanthocephalus bufonis*.

Location	A%	T%	C%	G%	AT%	AT-Skew	GC-Skew	Total
Whole mitochondrial genome	34.35	37.31	9.87	31.71	58.42	−0.28	0.53	14,056
Protein coding genes (PCGs)	18.88	38.07	9.27	33.78	56.95	−0.45	2.65	10,144
1st codon	21.58	30.12	9.58	38.72	51.70	−0.18	3.04	3383
2nd codon	13.07	47.97	10.68	28.28	61.05	−0.90	1.65	3381
3rd codon	21.98	36.12	7.54	34.35	58.11	−0.34	3.56	3380
tRNAs	25.20	37.48	10.50	26.82	62.68	−0.20	0.44	1238
rRNAs	28.18	35.54	10.61	25.68	63.72	−0.12	0.42	63.72
rrnS	27.97	37.06	8.74	26.22	65.03	−0.14	0.50	65.03
rrnL	28.30	34.58	11.78	25.33	62.89	−0.10	0.37	62.89
Non-coding region 1	28.36	33.61	19.34	18.69	61.97	−0.08	−0.02	610
Non-coding region 2	26.64	30.22	5.57	37.57	56.86	−0.06	0.74	503

**Table 6 animals-13-01256-t006:** Relative synonymous codon usage (RSCU) of *Pseudoacanthocephalus bufonis* (The asterisk represents termination codon).

Codon	aa	No.	%	Codon	aa	No.	%
TAA	*	6	0.18	TTA	Leu	183	5.44
TAG	*	3	0.09	TTG	Leu	188	5.58
GCA	Ala	48	1.43	AAA	Lys	34	1.01
GCC	Ala	16	0.48	AAG	Lys	12	0.36
GCG	Ala	20	0.59	ATA	Met	86	2.55
GCT	Ala	50	1.49	ATG	Met	90	2.67
CGA	Arg	10	0.30	TTC	Phe	28	0.83
CGC	Arg	3	0.09	TTT	Phe	186	5.52
CGG	Arg	9	0.27	CCA	Pro	27	0.80
CGT	Arg	18	0.53	CCC	Pro	14	0.42
AAC	Asn	9	0.27	CCG	Pro	8	0.24
AAT	Asn	49	1.46	CCT	Pro	23	0.68
GAC	Asp	10	0.30	AGA	Ser	36	1.07
GAT	Asp	43	1.28	AGC	Ser	20	0.59
TGC	Cys	16	0.48	AGG	Ser	108	3.21
TGT	Cys	60	1.78	AGT	Ser	55	1.63
CAA	Gln	9	0.27	TCA	Ser	20	0.59
CAG	Gln	11	0.33	TCC	Ser	4	0.12
GAA	Glu	23	0.68	TCG	Ser	5	0.15
GAG	Glu	53	1.57	TCT	Ser	45	1.34
GGA	Gly	36	1.07	ACA	Thr	18	0.53
GGC	Gly	28	0.83	ACC	Thr	12	0.36
GGG	Gly	304	9.03	ACG	Thr	12	0.36
GGT	Gly	114	3.39	ACT	Thr	39	1.16
CAC	His	10	0.30	TGA	Trp	29	0.86
CAT	His	38	1.13	TGG	Trp	106	3.15
ATC	Ile	19	0.56	TAC	Tyr	23	0.68
ATT	Ile	129	3.83	TAT	Tyr	108	3.21
CTA	Leu	35	1.04	GTA	Val	139	4.13
CTC	Leu	2	0.06	GTC	Val	41	1.22
CTG	Leu	51	1.51	GTG	Val	176	5.23
CTT	Leu	55	1.63	GTT	Val	205	6.09

## Data Availability

The mitogenome sequence of *Pseudoacanthocephalus bufonis* obtained in this study was deposited in the GenBank database. Voucher specimens of *P. bufonis* were deposited in the College of Life Sciences, Hebei Normal University, Hebei Province, under the accession number HBNU-A-2022A001L.

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
