# Peer review of "Phylomitogenomic Analyses Provided Further Evidence for the Resurrection of the Family Pseudoacanthocephalidae (Acanthocephala: Echinorhynchida)"

_animals, 2023, doi:10.3390/ani13071256_

Round 1

Reviewer 1 Report

Referee’s Comments on MS animals-2248601

The manuscript (MS) entitled " Phylomitogenomic analyses provided further evidence for resurrection of the family Pseudoacanthocephalidae (Acanthocephala: Echinorhynchida)" is an important paper which describes the mitochondrial genome of Pseudoacanthocephalus bufonis and provides further evidence for resurrection of the family Pseudoacanthocephalidae. The manuscript (MS) is well written, scientifically sound and the outcomes are well explained, and it is of interest to the readership of animals. Some minor comment listed as below.

Point 1: The current Abstract is vague with no introduction to importance of acanthocephalans.

Point 2: Keywords: Parasite should be deleted.

Point 3: The objectives of this study were not clearly presented. It is suggested that the authors clearly describe the objectives of the study in the last paragraph of the introduction.

Point 4: The method section should be better detailed and justified. First, the authors should explain why only 8,000 generations were executed. Second, why the maximum discrepancy 136 of the convergence result is 0.017 rather than 0.01.

Point 5: Do you use partitioning (different models for different genes of the mt genome)? Please clarify.

Point 6: I would suggest use standard tRNA gene name. such as trnQ.

Point 7: Make sure that your raw data and mt genome are submitted to GenBank. For now, the number your provide yields no result at all (even hidden)

Point 8: 3.6. Molecular Phylogeny. More in-depth analysis should be done to show how special or how useful the mt genome they obtain.

Point 9: lines 320-322 look like the conclusion that is somewhat redundant. I would recommand deleting them.

Point10: Table 3. This column of Gap or overlap needs to be aligned. Anticodon needs to be capitalized.

Point 11: All gene should be in italics. Such as line 207. Write whole the word "cox3, nad4 and nad6" in italic.

Point 12: The image quality of figure 7 should be improved.

Point 13: There are minor typo-errors throughout the MS, and should be corrected during revision. Such as Line 137. Add a dot after 0.017. 

Author Response

Reviewer 1

The manuscript (MS) entitled "Phylomitogenomic analyses provided further evidence for resurrection of the family Pseudoacanthocephalidae (Acanthocephala: Echinorhynchida)" is an important paper which describes the mitochondrial genome of Pseudoacanthocephalus bufonis and provides further evidence for resurrection of the family Pseudoacanthocephalidae. The manuscript (MS) is well written, scientifically sound and the outcomes are well explained, and it is of interest to the readership of animals. Some minor comment listed as below.

Point 1: The current Abstract is vague with no introduction to importance of acanthocephalans.

R: Ok, added “Acanthocephalans are of veterinary, medical and economic importance, due to cause disease in domestic animals, wildlife and humans”.

Point 2: Keywords: Parasite should be deleted.

R: Ok, done.

Point 3: The objectives of this study were not clearly presented. It is suggested that the authors clearly describe the objectives of the study in the last paragraph of the introduction.

R: Ok, added “In order to further test the monophyly of the orders Echinorhynchida and Polymor-phida, assess the validity of the recently resurrected family Pseudoacanthocephalidae and clarify the evolutionary relationships of the Pseudoacanthocephalidae and the other families in Palaeacanthocephala using mitogenomic data, the complete mitochondrial genome of P. bufonis (Shipley, 1903)…”

Point 4: The method section should be better detailed and justified. First, the authors should explain why only 8,000 generations were executed. Second, why the maximum discrepancy 136 of the convergence result is 0.017 rather than 0.01.

R: The calculation in this study has converged since the generation 2000, and the topology is consistent with that in the present paper. The results of operation with 8000 generations are stable and reliable and we also considered the consumption and limitation of computing resources. According to the recommendation of PhyloBayes, maximum diversity less than 0.1 can be considered as good convergence result. The two parameters for judging the tree convergence, included maximum diversity of the convergence result to be 0.017, and mean discrepancy to be 0.001. Thus three decimal places are reserved to make mean discrepancy not all 0. But mean diversity is not shown in the paper.

Point 5: Do you use partitioning (different models for different genes of the mt genome)? Please clarify.

R: Yes. As shown in Table 2, two partition schemes were only applied for ML method: full partition (FP) that provides the best-fitting model for each individual gene; and merged partition (MP) that implements a greedy strategy starting with the full partition model and subsequently merging pairs of genes until the model fit does not improve.

Point 6: I would suggest use standard tRNA gene name, such as trnQ.

R: Ok, revised.

Point 7: Make sure that your raw data and mt genome are submitted to GenBank. For now, the number your provide yields no result at all (even hidden)

R: You can retrieve the data under accession number MZ958236.1 in the Nucleotide database of NCBI or click the link directly: https://www.ncbi.nlm.nih.gov/nuccore/MZ958236.1

Point 8: 3.6. Molecular Phylogeny. More in-depth analysis should be done to show how special or how useful the mt genome they obtain.

R: Ok, revised.

Point 9: lines 320-322 look like the conclusion that is somewhat redundant. I would recommand deleting them.

R: Ok, deleted.

Point10: Table 3. This column of Gap or overlap needs to be aligned. Anticodon needs to be capitalized.

R: The alignment problem should be caused by typography. We adjusted that.

Point 11: All gene should be in italics. Such as line 207. Write whole the word "cox3, nad4 and nad6" in italic.

R: Ok, revised.

Point 12: The image quality of figure 7 should be improved.

R: The current format of figures is produced by the review-system of journal with compressed size. We resubmitted all of figures as separated files with large enough size.

Point 13: There are minor typo-errors throughout the MS, and should be corrected during revision. Such as Line 137. Add a dot after 0.017.

R: Ok, revised.

Reviewer 2 Report

This is an interesting and well constructed paper. It is useful to note that despite the problems morphologists have had in accepting the validity of the genus Pseudoacanthocephalus in the past, molecular evidence for the family is compelling.The authors should note that a mitochondrial genome sequence is singular and edit the text accordingly eg line 118... was deposited. Also could they please check that they have used the definite article where it is needed eg line125...information on the representatives, line 126 the Gnathostomulida. Is there more to be said about the GT & AT skew data (fig 3)? That the AT skew was an outlier seems to call for some comment

1 The main question addressed by the research is the phylogeny of the Pseudoacanthocephalidae

2.The topic is original and address a specific gap in the field.

3.This is the first mitogenome study of this family

new data is provided which adds to our understanding of the acanthocephalan genome and allows comparisons with other published results as to the phylogeny of the acanthocephalan in general and the Orders Polymorphida and Echinorhynchida in particular

4. The methods seemed to be appropriate for both the molecular and morphological work; The authors used methods that had been published previously to elucidate the mitogenome and generate phylogenetic trees. I personally can see no reason to suggest improvements. Reviewers who are actively involved in extraction and analysis of genomic DNA may be able to make critical comment on the methodology

5. The conclusions are consistent with the evidence and arguments presented and they address the main question.

6. There were no unnecessary self citations. Many of the relevant papers are more than 5 years old. It would be bad science to limit the analysis of and reference to relevant literature to the past 5 years. Important understandings of the phylogeny and relationships of the acanthocephalan were published more than 20 years ago and the authors are to be commended on their knowledge of the relevant literature

7. The text of Figures 6 and 7 is too small to read. Is this due to the formatting of the paper for review or were the figures too small as submitted

Author Response

Reviewer 2

Comments and Suggestions for Authors

This is an interesting and well constructed paper. It is useful to note that despite the problems morphologists have had in accepting the validity of the genus Pseudoacanthocephalus in the past, molecular evidence for the family is compelling. The authors should note that a mitochondrial genome sequence is singular and edit the text accordingly eg line 118... was deposited. Also could they please check that they have used the definite article where it is needed eg line125...information on the representatives, line 126 the Gnathostomulida. Is there more to be said about the GT & AT skew data (fig 3)? That the AT skew was an outlier seems to call for some comment.

R: We re-submitted the improved version of manuscript and checked all the information of representatives of ingroup and outgroup for the phylogeny. We also added some comments on the GT & AT skew data.

The text of Figures 6 and 7 is too small to read. Is this due to the formatting of the paper for review or were the figures too small as submitted

R: The current format of figures and tables is produced by the review-system of journal with compressed size. We resubmitted all of figures as separated files with large enough size.

Reviewer 3 Report

Review on Manuscript ID: animals-2248601

Phylomitogenomic analyses provided further evidence for resurrection of the family Pseudoacanthocephalidae (Acanthocephala: Echinorhynchida)

by Zhao and co-workers

The authors characterize the mitochondrial genome of the thorny-headed worm (Acanthocephala) Pseudoacanthocephalus (Acanthocephalus) bufonis, a representative of Pseudoacanthocephalidae (Palaeacanthocephala). In P. bufonis, the mitogenome spans roughly 14 kb. The plus strand includes 36 genes, i.e., 12 protein-coding, 22 tRNA, and 2 rRNA genes. Two non-coding regions and a strong GC-skew complete the picture. Phylogenetic analysis relied on translated sequences of concatenated the protein-coding genes upon pruning of ambiguously aligned positions. Maximum Likelihood and Bayesian Inference trees suggest that the traditional taxon Echinorhynchida is paraphyletic. The results further question the status of Polyacanthorhynchida as a separate order and support the resurrection of Pseudoacanthocephalidae as a distinct group with potentially closer phylogenetic relation to Cavisomatidae. In turn, Polymorphida are confirmed to be monophyletic, with Polymorphidae potentially being closely related to Centrorhynchidae than to Plagiorhynchidae.

In this reviewer’s opinion, the manuscript is overall well-written. Still, I have suggestions and edits which I recommend the authors to implement into a revised manuscript. The species status of the acanthocephalan under study appears particularly critical to this reviewer.

Major points

- Outline in the Materials & Methods how the micrographs were taken.

- List all referenced studies in Table 1 in the References at the end of the manuscript.

- Findings related to morphology and species determination (Figure 1) should be shifted to the Results & Discussion section. Structures in micrographs of Figure 1 ought to be labelled. A -> hooks or thorns; B -> presoma, metasoma or trunk, lemniscs, double-walled receptacle; C -> testicles, everted Bursa copulathrix, body cavity of trunk, tegument, or epidermis, subtegumental musculature etc.; D -> body cavity of trunk, tegument or epidermis, subtegumental musculature, uterus bell, vagina, sphincter, genital pore etc.; E -> shell membranes, acanthor, anterior pole. Obviously, there are several papers which could be considered in addition to your references 31-34 but this one (10.1046/j.1463-6395.2003.00143.x) might prove useful in respect to a standardized terminology.

- In the current draft, it is stated: “The specimens were identified as P. bufonis based on morphological features reported in previous studies [31–34].” To my understanding, this is not possible because of inconsistencies between your references and between your findings and the references. Thus, the studies you are referring to are Petrochenko (1953), Kennedy & Murray (1982), Wang (1989), and Bush et al. (2009). Yet, Kennedy & Murray (1982) delineate that Petrochenko (1953) referred to a different species for which they introduced an extra name (Acanthocephalus breviprostatus nom. nov.). According to this, Petrochenko (1953) does not include data on Pseudoacanthocephalus (Acanthocephalus) bufonis. Furthermore, the measures emerging from your Figure 1 are partially inconsistent with the amended species descriptions for P. bufonis by Kennedy & Murray (1982) and Bush et al. (2009). For example, the acanthor should measure 78 µm long at minimum according to Kennedy & Murray (1982), while it has 65 µm in your Figure 1E. If the acanthor is holoechinate as it should be according to Kennedy & Murray (1982), or not cannot be judged from the micrograph in your Figure 1E. Also, the acanthor is about half the length of the entire egg in Fig. 3 in the study by Bush et al. (2009) while it takes almost 60% of the egg length in your Fig. 1E. Moreover, the lemniscs in your Fig. 1B are about 500 µm long but they should have 800 up to 1,420 µm according to Bush et al. (2009). Finally, the referenced study by Wang (1989) was published in the Journal of Pujian Teachers University (Natural Science). Using this information, an internet search did not yield any useful result. The referenced study by Wang (1989) hence seems to be inaccessible. Please, explain why you feel sure about having sampled P. bufonis and not another species. In the latter case, you might consider adjusting corresponding phrases in the manuscript from P. bufonis to Pseudoacanthocephalus spec. – provided you confirm the allocation to this genus.

- In my opinion it is worthwhile to mention and briefly discuss (Results & Discussion) that the phylogenetic tree reconstruction confirms paraphyletic Eurotatoria, in particular a closer relation of bdelloid rotifers to acanthocephalans than to monogonont rotifers.

- Labels in Figures 2, 4, 5, 6, and 7 are much too small. Please, adjust.

- Figures 6 and 7 are generally much too small. Please, adjust.

Minor points

- Use italics for gene symbols.

- Legend to Figure 2 and header of Table 3: To my understanding, reference to plus and minus strands is pointless; all genes reside on the plus strand.

- Line 197: component -> composition?

- Line 115: Sentence makes no sense to me. Delete “of the base position”?

- Line 194+195: delete “, which include 3,358 codons”

- Line 212: sequences. -> sequences (Fig. 4; Table 5).

- Line 104: were -> was; line 109: -based based -> based; line 124: includes -> included; lines 129+130: eliminated -> pruned; line 137: 0.0017 The -> 0.0017. The; lines 141+142+224+303: excess free space characters; line 154: weak -> weakly; line 245: 62] There -> 62]. There

Author Response

Reviewer 3

Comments and Suggestions for Authors

Review on Manuscript ID: animals-2248601

Phylomitogenomic analyses provided further evidence for resurrection of the family Pseudoacanthocephalidae (Acanthocephala: Echinorhynchida)

The authors characterize the mitochondrial genome of the thorny-headed worm (Acanthocephala) Pseudoacanthocephalus (Acanthocephalus) bufonis, a representative of Pseudoacanthocephalidae (Palaeacanthocephala). In P. bufonis, the mitogenome spans roughly 14 kb. The plus strand includes 36 genes, i.e., 12 protein-coding, 22 tRNA, and 2 rRNA genes. Two non-coding regions and a strong GC-skew complete the picture. Phylogenetic analysis relied on translated sequences of concatenated the protein-coding genes upon pruning of ambiguously aligned positions. Maximum Likelihood and Bayesian Inference trees suggest that the traditional taxon Echinorhynchida is paraphyletic. The results further question the status of Polyacanthorhynchida as a separate order and support the resurrection of Pseudoacanthocephalidae as a distinct group with potentially closer phylogenetic relation to Cavisomatidae. In turn, Polymorphida are confirmed to be monophyletic, with Polymorphidae potentially being closely related to Centrorhynchidae than to Plagiorhynchidae. In this reviewer’s opinion, the manuscript is overall well-written. Still, I have suggestions and edits which I recommend the authors to implement into a revised manuscript. The species status of the acanthocephalan under study appears particularly critical to this reviewer.

R: We are very grateful for the useful comments of the referee, which have helped us to improve our manuscript. We have carefully revised the manuscript guided by these constructive comments. We provided the morphological characters and the ITS region of our specimens for exact species identification.

Major points

- Outline in the Materials & Methods how the micrographs were taken.

R: Ok, added.

- List all referenced studies in Table 1 in the References at the end of the manuscript.

R: Ok, revised.

- Findings related to morphology and species determination (Figure 1) should be shifted to the Results & Discussion section. Structures in micrographs of Figure 1 ought to be labelled. A -> hooks or thorns; B -> presoma, metasoma or trunk, lemniscs, double-walled receptacle; C -> testicles, everted bursa copulathrix, body cavity of trunk, tegument, or epidermis, subtegumental musculature etc.; D -> body cavity of trunk, tegument or epidermis, subtegumental musculature, uterus bell, vagina, sphincter, genital pore etc.; E -> shell membranes, acanthor, anterior pole. Obviously, there are several papers which could be considered in addition to your references 31-34 but this one (10.1046/j.1463-6395.2003.00143.x) might prove useful in respect to a standardized terminology.

R: We provided the morphological characters (including range of measurements) and added the comments on species identification of this species in the Results & Discussion section. We also indicated most of the above-mentioned structures suggested by the reviewer in the Figure 1, but the main objective of this study is not to re-describe the detailed morphology of this species. We also provided some additional published papers reported this species.

- In the current draft, it is stated: “The specimens were identified as P. bufonis based on morphological features reported in previous studies [31–34].” To my understanding, this is not possible because of inconsistencies between your references and between your findings and the references. Thus, the studies you are referring to are Petrochenko (1953), Kennedy & Murray (1982), Wang (1989), and Bush et al. (2009). Yet, Kennedy & Murray (1982) delineate that Petrochenko (1953) referred to a different species for which they introduced an extra name (Acanthocephalus breviprostatus nom. nov.). According to this, Petrochenko (1953) does not include data on Pseudoacanthocephalus (Acanthocephalus) bufonis. Furthermore, the measures emerging from your Figure 1 are partially inconsistent with the amended species descriptions for P. bufonis by Kennedy & Murray (1982) and Bush et al. (2009). For example, the acanthor should measure 78 µm long at minimum according to Kennedy & Murray (1982), while it has 65 µm in your Figure 1E. If the acanthor is holoechinate as it should be according to Kennedy & Murray (1982), or not cannot be judged from the micrograph in your Figure 1E. Also, the acanthor is about half the length of the entire egg in Fig. 3 in the study by Bush et al. (2009) while it takes almost 60% of the egg length in your Fig. 1E. Moreover, the lemniscs in your Fig. 1B are about 500 µm long but they should have 800 up to 1,420 µm according to Bush et al. (2009). Finally, the referenced study by Wang (1989) was published in the Journal of Pujian Teachers University (Natural Science). Using this information, an internet search did not yield any useful result. The referenced study by Wang (1989) hence seems to be inaccessible. Please, explain why you feel sure about having sampled P. bufonis and not another species. In the latter case, you might consider adjusting corresponding phrases in the manuscript from P. bufonis to Pseudoacanthocephalus spec. – provided you confirm the allocation to this genus.

R: (1) Although Kennedy (1982) proposed a new name Acanthocephalus breviprostatus nom. nov. for Petrochenko (1953)’s material of P. bufonis, this taxnomical change was not supported by subsequent taxonomical studies, i.e., Amin et al. (2008, 2014) and Amin (2013), which all accepted Acanthocephalus breviprostatus nom. nov. as a synonym of P. bufonis (Shipley, 1903). (2) In the revised version of manuscript, we also provided the morphological characters (range of measurements) of this species based on specimens collected from the spot-legged tree frog P. megacephalus in Hainan island, China. Moreover, we also sequenced the ITS region (OQ550505, OQ550506) of our specimens collected from P. megacephalus. Pairwise comparison of the ITS sequences of our specimens with the available ITS data (KC491878–KC491883) of P. bufonis reported by Tkach et al. (2013) displayed only 0.17% of nucleotide divergence. Thus we confirmed our specimens to be P. bufonis. In fact, we are preparing another paper about the morphological variation of this species from different hosts. We collected P. bufonis from five different frogs. Whilst there is a considerable intraspecific variation in the size, shape and armature of the proboscis (proboscis armed with 16–20 longitudinal rows of 2–6 rooted hooks each), the lengths of trunk, proboscis receptacle and lemnisci, and the size of eggs among different individuals. However, our genetic evidence proved that these specimens with considerable intraspecific variation in morphology all represented a single species (no nucleotide divergence detected in ITS and only 0.31–1.37% nucleotide divergence detected among different individuals collected from different hosts vs interspecific nucleotide divergence 13.4-28.7% in ITS and 27.4-28.9% in cox1, respectively). (3) Wang (1989) provided detailed morphological characters of different developed stages of P. bufonis. We also compared the morphometrics of our specimens with Wang (1989)’s data, that are more or less identical. 

- In my opinion it is worthwhile to mention and briefly discuss (Results & Discussion) that the phylogenetic tree reconstruction confirms paraphyletic Eurotatoria, in particular a closer relation of bdelloid rotifers to acanthocephalans than to monogonont rotifers.

R: Ok, added “Phylogenetic trees generated from BI and ML methods under different models have similar topologies and indicated that the Acanthocephala to be monophyletic, which was widely accepted in previous studies. However, the evolutionary relationships of the Acanthocephala and the three subtaxa of Rotifera (Monogononta, Bdelloidea, Seisonidea) is under debate for a long time [19, 71–73]. The present phylogenetic results showed that the Acanthocephala is sister to Bdelloidea (Rotaria rotatoria, Philodina citrina) and rejected the monophyly of Eurotatoria (Monogononta + Bdelloidea) and Pararotatoria (Seisonidea + Acanthocephala), which are identical to the previous phylogenetic results using EST li-braries [72] and mitogenomic data [19], but conflicted with some other phylogenies based on 18S rDNA and transcriptomic data [71, 73]”.

- Labels in Figures 2, 4, 5, 6, and 7 are much too small. Please, adjust.

- Figures 6 and 7 are generally much too small. Please, adjust.

R: The current format of figures (including labels) and tables is produced by the review-system of journal with compressed size. We resubmitted all of figures and tables as separated files with large enough size.

Minor points

- Use italics for gene symbols.

R: Ok, revised.

- Legend to Figure 2 and header of Table 3: To my understanding, reference to plus and minus strands is pointless; all genes reside on the plus strand.

R: Ok, deleted.

- Line 197: component -> composition?

R: Ok, revised.

- Line 115: Sentence makes no sense to me. Delete “of the base position”?

- Line 194+195: delete “, which include 3,358 codons”

R: Ok, revised.

- Line 212: sequences. -> sequences (Fig. 4; Table 5).

R: Ok, revised.

- Line 104: were -> was; line 109: -based based -> based; line 124: includes -> included; lines 129+130: eliminated -> pruned; line 137: 0.0017 The -> 0.0017. The; lines 141+142+224+303: excess free space characters; line 154: weak -> weakly; line 245: 62] There -> 62]. There

R: Ok, revised.

Round 2

Reviewer 3 Report

Review on revised draft (v1) of

Phylomitogenomic analyses provided further evidence for resurrection of the family Pseudoacanthocephalidae (Acanthocephala: Echinorhynchida)

by Zhao et al.

This reviewer recognizes improvements in the manuscript. Still, labelling of the Figures remains problematic. New labeling in Figure 1 is almost unreadable due to low contrast with background (white should be black), and labels in Figures 2 and 6 remain to be too small. Obviously, these are minor points which can be easily fixed. Please, see below for two major points which a revised draft should address.

i) Taxonomic assignment: The taxonomic assignment of the acanthocephalans studied by Zhao et al. to Pseudoacanthocephalus bufonis remains challenging. Probably due to my earlier comment, Zhao et al. have adjusted the wording. Instead of stating that the descriptions in various reference studies match the data on the acanthocephalans analyzed by themselves, it is now said that the measures more or less correspond. This does not adequately address the issue. Indeed, some of the measures by Zhao et al. differ considerably from the data provided by other authors, as illustrated by the following examples (more examples are included in the previous review):

- Number of proboscis hooks per row: The probosicis of “P. bufonis has twenty longitudinal rows (sometimes 18-20) with a row of 7 hooks alternating with two rows of 6 hooks” according to Petrochenko (1958: Acanthocephala of domestic and wild animals). Likewise, Shipley (1903) reports “6-8 rings, alternatively arranged with 14-16 longitudinal rows.” In contrast, Zhao et al. observed “16-20 longitudinal rows of 3-5 rooted hooks each.”

- Relative position of proboscis: Shipley (1903) states “The most conspicuous feature in which this species differs from the majority of its congeners is that the proboscis or introvert is not median and terminal, but projects from the trunk a little way, sometimes more, sometimes less, from the anterior end; it usually slopes forward, but it may stand out at right angles to the axis of the body like the handle of a walking-stick.” This is just not the case in the specimens shown in Figure 1 in the study by Zhao et al.

- Extension of receptacle and lemniscs: In the drawings by Yuen & Fernando (1967) the lemniscs have double the extension of the receptacle. Contrary to that, the lemniscs have about the same length as the receptacle in Figure 1B by Zhao et al.

My urgent recommendation is to delineate that the measures by Zhao et al. partially differ from the ones in the referenced studies. After having reflected on the differences the authors may argue why they nevertheless believe to have collected P. bufonis.

ii) Phylogenetic position of Seisonidea: In Figure 7, the branch to Seison spec splits from the lineage to Monogononta, Bdelloidea and Acanthocephala. From this, the authors infer that their results reject monophyletic Pararotatoria. However, such conclusion does not take into account recent findings, according to which a basal braching of Seisonidea inside Syndermata results from long branch attraction. Indeed, Mauer et al. (2021; DOI: 10.1186/s12864-021-07857-y) demonstrated that, when controlling for long branch attraction, there is a regrouping inside Syndermata in sequence-based tree reconstruction, from a basal branching of the Seisonidea to a sister-group relationship with Acanthocephala. Thus, they received monophyletic Pararotatoria when avoiding long branch attraction. Before that, Sielaff et al. (2016; DOI: 10.1016/j.ympev.2015.11.017) had carved out that misleading signal (branch length heterogeneity, compositional heterogeneity etc.) promotes a basal branching of Seisonidea in sequence-based tree reconstruction, while mitochondrial gene order supports monophyletic Pararotatoria. In my opinion, the position of Seisonidea does not need to be addressed in the work of Zhao et al. But if the authors decide to do so then a balanced discussion of the topic is indicated. In the current draft, this is not the case.

Minor point

-          Lines 254/255: misspelling

Author Response

Detailed responses to comments

Phylomitogenomic analyses provided further evidence for resurrection of the family Pseudoacanthocephalidae (Acanthocephala: Echinorhynchida)

This reviewer recognizes improvements in the manuscript. Still, labelling of the Figures remains problematic. New labeling in Figure 1 is almost unreadable due to low contrast with background (white should be black), and labels in Figures 2 and 6 remain to be too small. Obviously, these are minor points which can be easily fixed.

R: revised.

Please, see below for two major points which a revised draft should address.

  1. i) Taxonomic assignment: The taxonomic assignment of the acanthocephalans studied by Zhao et al. to Pseudoacanthocephalus bufonis remains challenging. Probably due to my earlier comment, Zhao et al. have adjusted the wording. Instead of stating that the descriptions in various reference studies match the data on the acanthocephalans analyzed by themselves, it is now said that the measures more or less correspond. This does not adequately address the issue. Indeed, some of the measures by Zhao et al. differ considerably from the data provided by other authors, as illustrated by the following examples (more examples are included in the previous review):

- Number of proboscis hooks per row: The probosicis of “P. bufonis has twenty longitudinal rows (sometimes 18-20) with a row of 7 hooks alternating with two rows of 6 hooks” according to Petrochenko (1958: Acanthocephala of domestic and wild animals). Likewise, Shipley (1903) reports “6-8 rings, alternatively arranged with 14-16 longitudinal rows.” In contrast, Zhao et al. observed “16-20 longitudinal rows of 3-5 rooted hooks each.”

- Relative position of proboscis: Shipley (1903) states “The most conspicuous feature in which this species differs from the majority of its congeners is that the proboscis or introvert is not median and terminal, but projects from the trunk a little way, sometimes more, sometimes less, from the anterior end; it usually slopes forward, but it may stand out at right angles to the axis of the body like the handle of a walking-stick.” This is just not the case in the specimens shown in Figure 1 in the study by Zhao et al.

- Extension of receptacle and lemniscs: In the drawings by Yuen & Fernando (1967) the lemniscs have double the extension of the receptacle. Contrary to that, the lemniscs have about the same length as the receptacle in Figure 1B by Zhao et al.

My urgent recommendation is to delineate that the measures by Zhao et al. partially differ from the ones in the referenced studies. After having reflected on the differences the authors may argue why they nevertheless believe to have collected P. bufonis.

R: (i) Van Cleave (1937) firstly described Pseudoacanthocephalus bufonis from China and reported the proboscis armed with 15-19 longitudinal rows of 4-6 hooks each. Subsequently, Wang (1989) redescribed P. bufonis based on specimens collected from Bufo melanostictus in Chian, which has proboscis with 16-18 longitudinal rows of 5 hooks each. Bush et al. (2009) redescribed P. bufonis based on material collected from 8 frogs in China, including Polypedates megacephalus Hollowell, the same host as the present specimens collected. Bush et al. (2009) reported the proboscis with 16-18 longitudinal rows of 5-6 hooks each in their specimens of P. bufonis. They also commented on Shipley’s (1903) description and considered ‘‘6–8 alternating rings.’’ in Shipley’s (1903) description would be equivalent to 3–4 hooks in each longitudinal row. In the present study, we observed proboscis with 16-20 longitudinal rows of 3-5 hooks each. Consequently, when large numbers of specimens collected from different hosts and localities were observed, P. bufonis displayed a considerable intraspecific morphological variation in the armature of the proboscis. Moreover, Tkach et al. (2013) sequenced the ITS region (KC491878–KC491883) of P. bufonis based on specimens collected by Bush et al. (2009), also including the material collected from Polypedates megacephalus, which only showed 0.17% of nucleotide divergence between Tkach et al.’s (2013) material of this species and our present specimens collected from P. megacephalus in the ITS region. Thus we confirmed our specimens and Tkach et al.’s (2013) material to be congeneric, both belonging to P. bufonis.

(ii) Some published papers proved that the relative length of lemniscs and receptacle is not a stable feature for species identification of acanthocephalans. The length of lemnisci is related to the body size, developmental stage and hosts of acanthocephalans. For example, Amin et al. (2008) reported that specimens of Pseudoacanthocephalus nguyenthileae collected from Rana guentheri with large size of trunk, have lemnisci distinctly longer than proboscis receptacle, but specimens of P. nguyenthileae collected from Paa verrucospinosa with small size of trunk, have lemnisci shorter than proboscis receptacle. Wang (1989) studied the life history of P. bufonis and described the morphology of different developmental stage of this species, which clearly showed the variation of relative length of lemnisci and receptacle with different developmental stage of worm. In the drawings by Petrochenko (1953, 1958) and Kennedy (1982), the lemnisci is only slightly longer than receptacle.

(iii) The relative position of proboscis may be artificial when fixed specimens. We don’t think it is valuable for species identification. Additionally, some previous studies (Li et al. 2019, parasitology, 1-8; Li et al. 2017, Parasitology International, 693–698) indicated that we seriously underestimated the range/level of intraspecific morphological variation. The validity of many morphospecies of acanthocephalans will be challenged by genetic data in the future.

  1. ii) Phylogenetic position of Seisonidea: In Figure 7, the branch to Seison spec splits from the lineage to Monogononta, Bdelloidea and Acanthocephala. From this, the authors infer that their results reject monophyletic Pararotatoria. However, such conclusion does not take into account recent findings, according to which a basal braching of Seisonidea inside Syndermata results from long branch attraction. Indeed, Mauer et al. (2021; DOI: 10.1186/s12864-021-07857-y) demonstrated that, when controlling for long branch attraction, there is a regrouping inside Syndermata in sequence-based tree reconstruction, from a basal branching of the Seisonidea to a sister-group relationship with Acanthocephala. Thus, they received monophyletic Pararotatoria when avoiding long branch attraction. Before that, Sielaff et al. (2016; DOI: 10.1016/j.ympev.2015.11.017) had carved out that misleading signal (branch length heterogeneity, compositional heterogeneity etc.) promotes a basal branching of Seisonidea in sequence-based tree reconstruction, while mitochondrial gene order supports monophyletic Pararotatoria. In my opinion, the position of Seisonidea does not need to be addressed in the work of Zhao et al. But if the authors decide to do so then a balanced discussion of the topic is indicated. In the current draft, this is not the case.

R: We accepted the reviewer’s suggestion and only discussed the monophyly of Eurotatoria (Monogononta + Bdelloidea) in the present manuscript.

Minor point

-Lines 254/255: misspelling

R: revised.